# Tumor microenvironment predicts local tumor extensiveness in PD-L1 positive nasopharyngeal cancer

Soehartati A. Gondhowiardjo[1], Handoko[1]*, Marlinda Adham[2], Lisnawati Rachmadi[3], Henry Kodrat[1], Demak Lumban Tobing[4], I. Made Haryoga[1], Agustinus Gatot Dwiyono[1], Yoseph Adi Kristian[1], Tiara Bunga Mayang Permata[1]

1 Department of Radiotherapy, Faculty of Medicine, Universitas Indonesia / Cipto Mangunkusumo National General Hospital, Jakarta, Indonesia, 2 Department of Ear, Nose and Throat–Head and Neck Surgery, Faculty of Medicine, Universitas Indonesia / Cipto Mangunkusumo National General Hospital, Jakarta, Indonesia, 3 Department of Anatomical Pathology, Faculty of Medicine, Universitas Indonesia / Cipto Mangunkusumo National General Hospital, Jakarta, Indonesia, 4 Department of Clinical Pathology, Dharmais National Cancer Hospital, Jakarta, Indonesia

* handoko45@yahoo.com

**Data Availability Statement:** All relevant data are within the paper and its Supporting Information files.

## Abstract

Tumor microenvironment have been implicated in many kind of cancers to hold an important role in determining treatment success especially with immunotherapy. In nasopharyngeal cancer, the prognostic role of this immune cells within tumor microenvironment is still doubtful. We conducted a study that included 25 nasopharyngeal cancer biopsy specimens to seek a more direct relationship between tumor infiltrating immune cells and tumor progression. Apart from that, we also checked the PD-L1 protein through immunohistochemistry. The PD-L1 was positively expressed in all our 25 samples with nasopharyngeal cancer WHO type 3 histology. Majority samples have >50% PD-L1 expression in tumor cells. We also found that denser local tumor infiltrating immune cells population have relatively much smaller local tumor volume. The inverse applied, with the mean local tumor volumes were $181.92 \text{ cm}^3 \pm 81.45 \text{ cm}^3$, $117.13 \text{ cm}^3 \pm 88.72 \text{ cm}^3$, and $55.13 \text{ cm}^3 \pm 25.06 \text{ cm}^3$ for mild, moderate, and heavy immune cells infiltration respectively (p = 0.013). Therefore, we concluded that tumor infiltrating immune cells play an important role in tumor progression, hence evaluating this simple and predictive factor may provide us with some valuable prognostic information.

## Introduction

The ability of mutated cells to escape immune recognition is one of the major mechanism of tumor cells to become malignant. It takes many years from a precancerous lesion to transform into cancerous lesion. Multiple ongoing and continuous mutations eventually drive some clones of mutated cells to acquire the ability to escape immune recognition. The tumor cells able to escape immune surveillance by various mechanism. One of the mechanism of immune escape by tumor cells that is well studied is through expression of various immune checkpoint

**Funding:** The grant for this study was received by SAG from Directorate of Research and Community Engagement, Universitas Indonesia. The funder played no role in study design, data collection, analysis, decision to publish or preparation of the manuscript. URL of funder: https://research.ui.ac.id/research/en/home-en/

**Competing interests:** The authors have declared that no competing interests exist.

molecules. The interplay between the presence of various immune checkpoint molecules between host immune cells and tumor cells result in downplay of inflammatory signals.[1]

Nasopharyngeal cancer, which is tightly related with Epstein-Barr Virus (EBV) infection, has a distinct way from escaping immune surveillance. There are various viral products expressed in host cells that resulted in expansion of immunosuppressive cells, evasion of immune recognition by host major histocompatibility complex, and downregulation of various pro-inflammatory cytokines.[2–5] Apart from that, EBV viral proteins have also been implicated in upregulation of immune checkpoint molecule such as Programmed Cell Death Ligand 1 (PD-L1).[6] The expression of PD-L1 results in further downplay of immune attack, thereby it facilitates resistance free tumor progression.

Immune cells within the milieu of cancer cells have been shown to be associated with patient's prognosis in various types of cancer.[7–9] Various evidence suggesting a better prognosis in a rich infiltrating lymphocytes tumor microenvironment in nasopharyngeal cancer. [10,11] However, there are also various reports suggesting a doubtful to a non-prognostic role of immune cells within nasopharyngeal tumor microenvironment.[12–14] Due to this conflicting evidence observed in nasopharyngeal cancer, there is a need to study the role of tumor infiltrating immune cells in a more straight forward manner with a more direct endpoint, thus eliminating many other confounding factors. In this study, we reported the relationship between local tumor infiltrating immune cells and local tumor volume.

## Method

### Patients selection and recruitment

Twenty-five microscopically confirmed nasopharyngeal cancer patients were recruited consecutively from April to October 2019 from ENT oncology clinic in Cipto Mangunkusumo General Hospital, Jakarta. All patients were aged 18 and above. They all gave written consent to participate in this study. All patients underwent pathological reconfirmation by a senior pathologist and pre-treatment imaging. The pre-treatment imaging to asses local and regional tumor extension was either nasopharyngeal CT Scan or MRI Scan with scan area from at least the level of frontal sinus until supraclavicular. Chest X-Ray, bone scan, and abdominal ultrasound were conducted to assess metastatic disease. In case of doubtful lesion, additional imaging such as thoracic CT-Scan or abdominal CT-Scan or MRI was carried out for confirmation. All patients enrolled in this study did not have any prior cancer treatment. They also did not have any other severe diseases such hematological diseases or infectious diseases. They all had a routine peripheral blood count checked within a week prior from nasopharyngeal biopsy.

### Immunohistochemistry (IHC) staining and evaluation

The process of PD-L1 IHC staining was performed in the Laboratory of Anatomical Pathology, Faculty of Medicine, Universitas Indonesia. The paraffin embedded tissue of those confirmed NPC patients were cut until thickness of 3 μm. It was then deparaffinized with Xylol, rehydrated with alcohol 100%, 96%, and 70%. The slides were then pre-treated with Tris-EDTA on pH 9 in decloaking chamber at 94˚C. It was further rinsed with Phosphate Buffer Saline (PBS), then stained with primary immunohistochemistry PD-L1 antibody (GeneTex: GTX104763). After stained with primary antibody, it was re-rinsed with PBS and counterstained with hematoxylin stain. Next, lithium carbonate solution was applied to the slides, then the slides were rehydrated again in alcohol 80%, 96%, and 100%. The stained was then cleared with xylol again and finally they were covered with glass slides.

Evaluation of PD-L1 staining was carried out based on the percentage of tumor cells that was stained positive on its cytomembrane. The percentage of positively stained tumor cells was

then categorized into 5 scores, with score 0: $\leq 5\%$; score 1: $6 \leq 25\%$; score 2: 26 to $\leq 50\%$; score 3: $>50\%$ of PD-L1 positivity in tumor cells. The intensity of staining was not scored. Any intensity of staining as long as presence was calculated as positive stained cells. All the stained slides of PD-L1 IHC were evaluated until at least 10 fields of 400 times microscopic magnification independently by two different pathologists. In case of different scoring results by two pathologists, a consensus on final scoring was made after thorough review done together.

### Tumor microenvironment evaluation

Tumor microenvironment in the context of this study was defined as tumor infiltrating immune cells. It included all kind of immune cells. The evaluation of tumor infiltrating immune cells was based on a thin cut of a 3 μm paraffin embedded tissue biopsy specimen which was stained with hematoxylin eosin (HE) dye. The HE stained slides were evaluated for presence of nasopharyngeal cancer cells. Then the infiltrating immune cells surrounding the tumor cells were evaluated in 10 different representative fields under a 400 times magnification microscope independently by two different pathologists. The similar method as carried out in evaluation of IHC scoring was used in case of differences in initial independent scoring evaluation. The infiltrating immune cells for each patient's biopsy specimen were categorized and scored into 3 scores (1 = mild infiltration, 2 = moderate infiltration, 3 = heavy infiltration).

### Tumor volume determination

Tumor volume was determined based pre-treatment 3D imaging, from either nasopharyngeal CT scan or MRI scan with contrast. The selected imaging was the one nearest to the date of nasopharyngeal tumor biopsy, with none more than 2 months apart. The tumor volume was determined by delineating the tumor in Eclipse™ treatment planning system by 2 different radiation oncologists. The contours were done separately for local or primary tumor volume and nodal tumor volume from each patient. The final tumor contours were decided by consensus between those 2 radiation oncologists, then the tumor volumes were recorded.

### Statistical analysis

There were 3 groups of results which was stratified based on tumor infiltrating immune cells. The descriptive statistical analysis was done for all 3 groups. The figure following ± was denoted as standard deviation in this article, unless stated differently. The ANOVA statistical test was used to compare the means of those 3 groups. A p value of less than 0.05 was determined as statistical significance. The statistical test was carried out with SPSS software version 25.0.

## Result

### Patients and samples characteristics

Twenty-five nasopharyngeal cancer patients were recruited. There were 5 (20%), 11 (44%), and 9 (36%) patients in mild, moderate, and heavy tumor infiltrating immune cells category, respectively. The H&E samples of different scores of tumor infiltrating immune cells were found in **Fig 1A–1C**. There was almost equal number of patients between age ≤50 years old and >50 years old among the whole group. There were more males 19 (76%) than female in the whole group. The majority of patient was in stage IVA locally advanced stage. (see **Table 1**) All patients had nasopharyngeal carcinoma WHO type 3 histology.

All 25 biopsy samples were positively stained with IHC PD-L1 antibody. There was no sample with IHC PD-L1 score of 0 or <5% positivity of cytomembranic PD-L1 staining. All

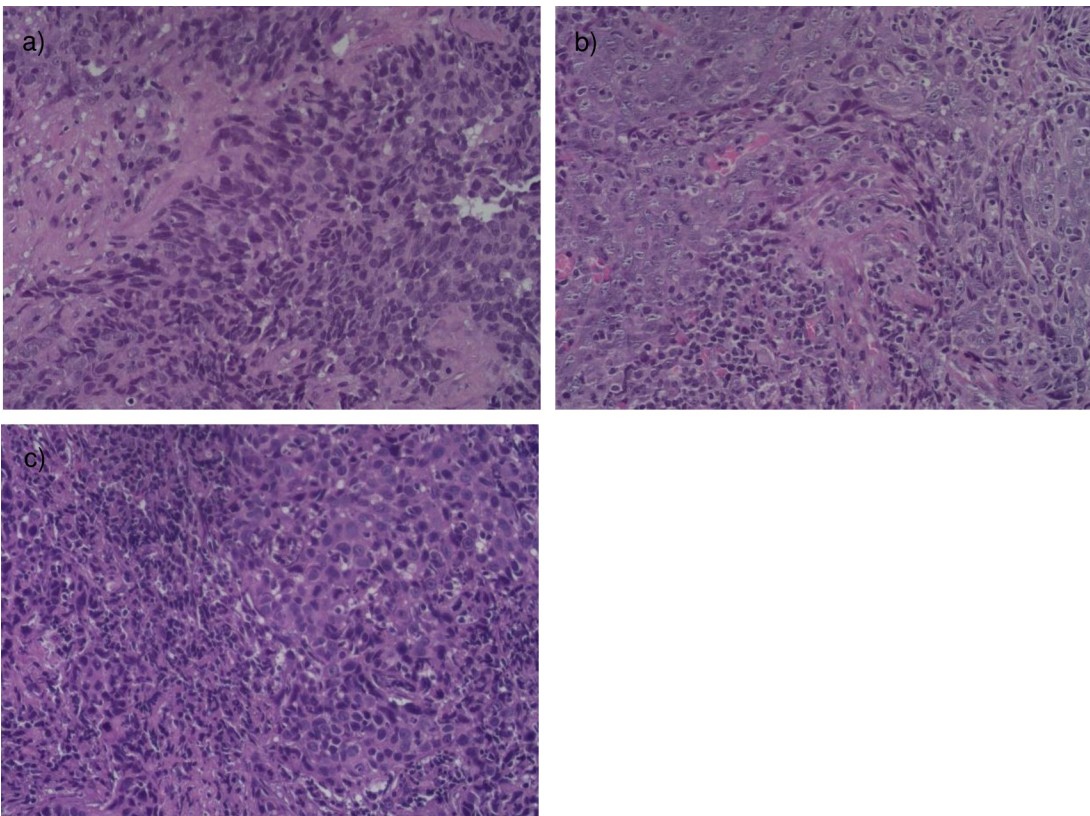

**Fig 1.** Histopathology slide stained with Hematoxylin Eosin dye in 400 times magnification showing density of tumor infiltrating immune cells nearby tumor cells in nasopharyngeal cancer a) score 1 = mild infiltration; b) score 2 = moderate infiltration; c) score 3 = heavy infiltration.

samples from the whole group had at least above 6% positivity of cytomembranic PD-L1 staining, with majority had more than 50% positivity in PD-L1 staining. There were 2, 8, and 15 samples with IHC PD-L1 score of 1, 2, and 3 respectively. However, there was no relationship found between positivity of PD-L1 cytomembranic staining with tumor infiltrating immune

**Table 1. Patient's characteristics segregated based on tumor infiltrating immune cells density score.**

| | Number of patients based on tumor infiltrating immune cells density score (percentage of total) | | | |
|---|---|---|---|---|
| **Variables** | **Mild** | **Moderate** | **Heavy** | **Total** |
| **Age** | | | | |
| ≤50 years old | 2 (8%) | 6 (24%) | 5 (20%) | 13 (52%) |
| >50 years old | 3 (12%) | 5 (20%) | 4 (16%) | 12 (48%) |
| **Gender** | | | | |
| Male | 3 (12%) | 10 (40%) | 6 (24%) | 19 (76%) |
| Female | 2 (8%) | 1 (4%) | 3 (12%) | 6 (24%) |
| **Stadium** | | | | |
| III | 0 (0%) | 2 (8%) | 2 (8%) | 4 (16%) |
| IVA | 1 (4%) | 7 (28%) | 5 (20%) | 13 (52%) |
| IVB | 4 (16%) | 2 (8%) | 2 (8%) | 8 (32%) |
| Total | 5 (20%) | 11 (44%) | 9 (36%) | 25 (100%) |

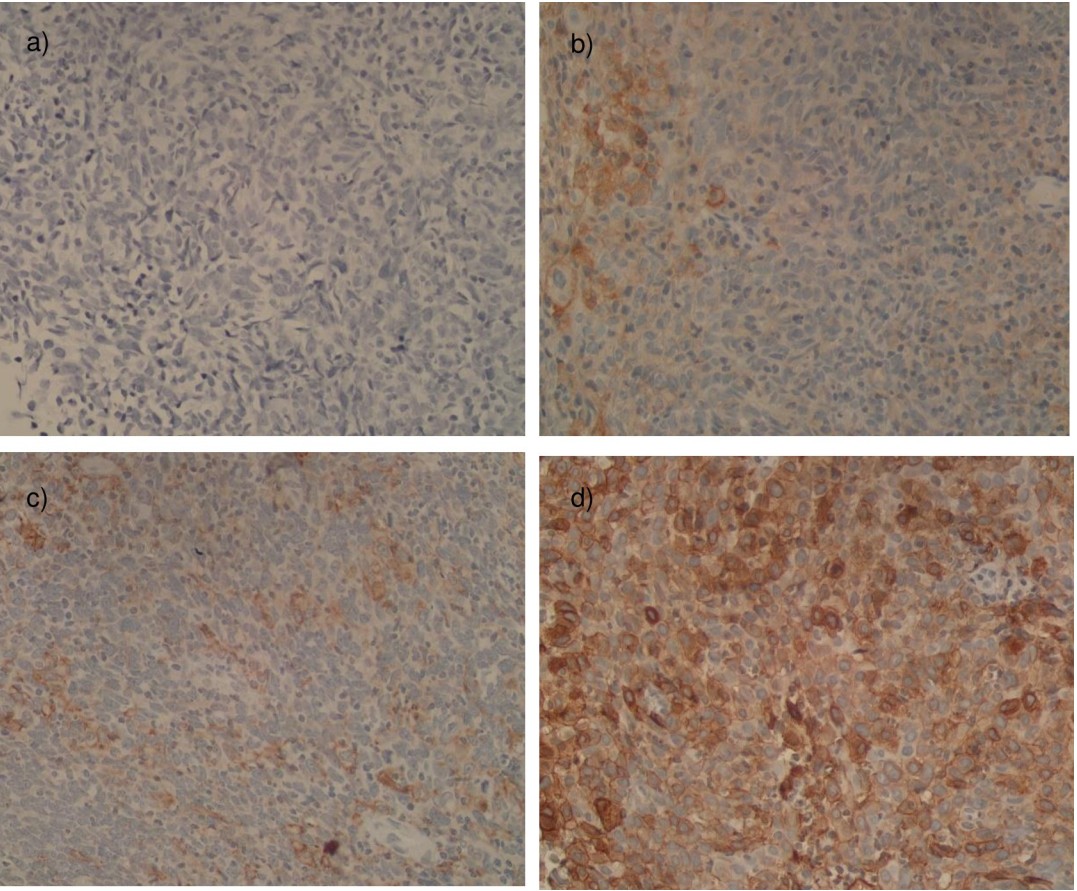

**Fig 2.** PD-L1 immunohistochemistry staining in 400 times magnification in nasopharyngeal cancer specimen a) Negative Control slide–no PD-L1 staining was observed on tumor cells; b) IHC PD-L1 in tumor cytomembranic cells—Score 1; c) IHC PD-L1 in tumor cytomembranic cells—Score 2; d) IHC PD-L1 in tumor cytomembranic cells—Score 3.

cells scores in our samples. Several samples of different IHC PD-L1 staining were found in **Fig 2A–2D**.

### Relationship between tumor volume, tumor infiltrating immune cells and systemic immune cells

There was an inverse relationship between local tumor infiltrating immune cells and local tumor volume. The mean local tumor volumes for each local tumor infiltrating immune cells score group were 181.92 cm$^3$ ± 81.45 cm$^3$, 117.13 cm$^3$ ± 88.72 cm$^3$, and 55.13 cm$^3$ ± 25.06 cm$^3$ for mild, moderate, and heavy infiltration score respectively (p = 0.013). In relatively larger local tumor volume patients, the tumor infiltrating immune cells were found to be sparser. The inverse was true with smaller local tumor volume patients having a denser tumor infiltrating immune cells. See **Fig 3**.

There was no relationship exist between local tumor infiltrating immune cells with nodal tumor volume and systemic immune cells. The mean nodal tumor volumes for each local tumor infiltrating immune cells score group were 56.76 cm$^3$ ± 39.67 cm$^3$, 104.71 cm$^3$ ± 163.29 cm$^3$, and 92.50 cm$^3$ ± 162.27 cm$^3$ for mild, moderate, and heavy infiltration score respectively (p = 0.838). The mean systemic leukocyte counts for those 3 group of patients segregated based on local tumor infiltrating immune cells score from 1 to 3 were 9,670 cells/μl ± 3,520

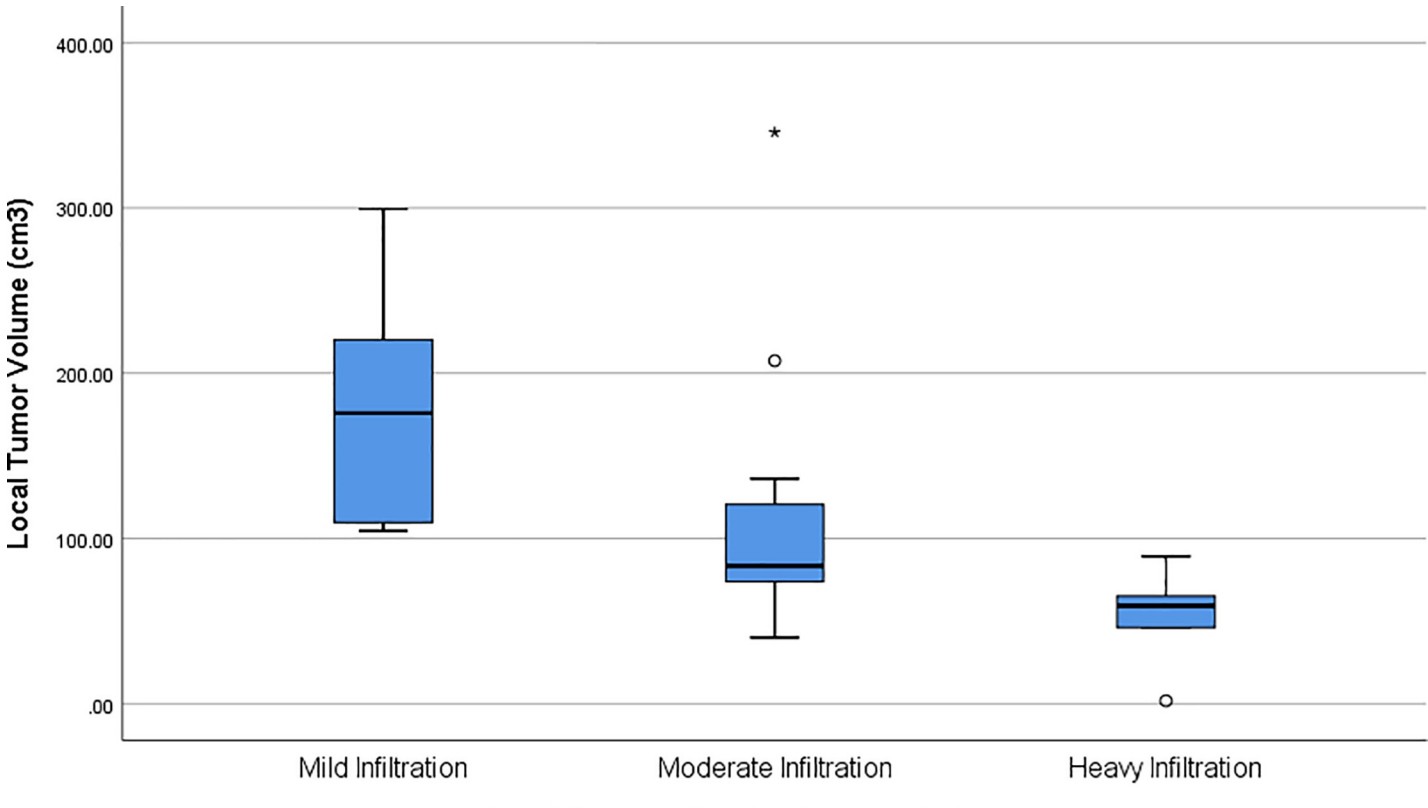

**Fig 3. Boxplot showing distribution and mean value of local tumor volume in nasopharyngeal cancer samples segregated based on local tumor infiltrating immune cells score density.**

cells/μl, 9,210 cells/μl ± 2,341 cells/μl, and 8,885 cells/μl ± 3,440 cells/μl, respectively (p = 0.896). There was also no relationship exist between all the leukocyte subtypes from basophil, eosinophil, neutrophil, lymphocyte, and monocyte among all the 3 groups of patients that were segregated based on local tumor infiltrating immune cells score.

## Discussion

The PD-L1 positivity in nasopharyngeal cancer is common especially for those with EBV infection or WHO type 3 undifferentiated carcinoma histology.[13] The high expression of PD-L1 in nasopharyngeal cancer has been shown in some studies to confer favorable prognosis to overall survival or progression free survival.[12,15] Nevertheless, there were also multiple studies shown that PD-L1 expression was not prognostic[12,13] or even conferred negative prognosis[16–19] in nasopharyngeal cancer. A recent meta-analysis confirmed that there was conflicting results for the prognostic role of PD-L1 expression in nasopharyngeal cancer.[20]

There were various factors that might affect the conflicting evidence of this prognostic role of PD-L1 in nasopharyngeal cancer. The antibody assay to assess the PD-L1 expression and the scoring method to evaluate extensiveness of PD-L1 expression were very variable.[21] The endpoint measured such as overall survival and disease free survival were indeed an important measure of prognosis, however, those endpoints were affected by too many other factors. Probably, it would be necessary to look at a smaller scope and a more direct relationship for instance PD-L1 expression and the local tumor microenvironment, among other factors.

Tumor microenvironment has been shown to be an important determinant of treatment success, thus affecting patient's prognosis.[10,22]

In a complex and heterogeneous tumor, where the tumors were constantly evolving, the dynamic interaction between host immune cells and tumor cells were important. The PD-L1 expression in the primary tumor site could be different with the PD-L1 expression in other sites such as metastatic sites due to different local microenvironment.[23] In order to better determine prognosis, a single biomarker such as PD-L1 may not be sufficient. A combination of multiple factors including patient's characteristics, tumor microenvironment, and possibly some other genomic or molecular parameters would better predict prognosis and guide better treatment options.

In our study, we showed that local tumor infiltrating immune cells was prognostic for predicting local tumor extensiveness. We assessed the whole infiltrating immune cells, instead of just specific immune cells because all kind of immune cells play an important and integral role in tumor resistance by immune cells. Immune cells were composed of components of innate immunity and adaptive immunity, which together are required to elicit a full blown immune response. Adaptive immunity such as T cells with all its subsets have been known to play a major and critical role in resisting cancer cells growth by exerting direct cytotoxic effects.[24] However, T cells alone were not enough to suppress tumor growth, all the other components of innate immunity were required to assist in recognizing, recruiting, activating, and enhancing the effect of cytotoxic T cells.[25]

Components of innate immunity such as neutrophils, NK cells, dendritic cells, macrophages, and all the cytokines produced by those cells have been shown to hold an important role in cancer immunotherapy.[10,22,25–30] A recent study of tumor infiltrating neutrophils in colorectal cancer patients indicated a higher density of neutrophils surrounding the tumor cells was associated with better overall prognosis and better response toward chemotherapy. [27] All those evidence suggesting that evaluating the whole tumor infiltrating immune cells might have an important role in predicting patient's prognosis and selecting the treatment especially the novel immunotherapy.

Although local tumor infiltrating immune cells was only predictive for local tumor extensiveness only, it might still hold a role in predicting overall patient's prognosis if various other potentially prognostic factors were incorporated. This study also pointed out that utilizing a quite simple and inexpensive method of evaluating overall tumor infiltrating immune cells might also work. This method might serve as the basis for reporting tumor infiltrating immune cells in other trials. Furthermore, this finding could trigger further exploratory work on tumor and immune cells interaction, especially for nasopharyngeal cancer.

In case of systemic immune cells, this study did not find any relationship among different scores of local tumor infiltrating immune cells from various systemic immune cells indices. This implied that the number of circulating systemic immune cells was not directly related to density of local tumor infiltrating immune cells. There were data indicating that failure of immune cells trafficked to tumor microenvironment led to failure of treatment including immunotherapy.[31] Nevertheless, there were studies reported prognostic role of circulating lymphocyte and neutrophil to immunotherapy response.[32,33] Therefore, a more thorough understanding is required to study and explore the role of systemic immune cells in affecting treatment response and finding ways to potentially increase T cells homing to tumor microenvironment.

## Conclusion

The local tumor infiltrating immune cells were related to local tumor extensiveness in PD-L1 positive nasopharyngeal cancer patients, with denser infiltration resulted in smaller local

tumor volume. Evaluating overall tumor infiltrating immune cells might be important and could serve as a simple and predictive marker for determining prognosis and possibly guiding treatment.

## Supporting information

**S1 Dataset.**
(XLSX)

## Acknowledgments

The authors would like to thanks dr. Freciyana Boedijono, an anatomical pathologist who have helped in evaluating IHC and tumor infiltrating immune cells from the samples of this study.

## Author Contributions

**Conceptualization:** Soehartati A. Gondhowiardjo, Agustinus Gatot Dwiyono.

**Data curation:** Handoko, I. Made Haryoga, Agustinus Gatot Dwiyono, Yoseph Adi Kristian.

**Formal analysis:** Handoko.

**Funding acquisition:** Soehartati A. Gondhowiardjo.

**Investigation:** I. Made Haryoga, Agustinus Gatot Dwiyono, Yoseph Adi Kristian.

**Methodology:** Soehartati A. Gondhowiardjo.

**Project administration:** I. Made Haryoga, Agustinus Gatot Dwiyono, Yoseph Adi Kristian.

**Resources:** Marlinda Adham, Lisnawati Rachmadi, Demak Lumban Tobing.

**Supervision:** Soehartati A. Gondhowiardjo, Marlinda Adham, Lisnawati Rachmadi, Henry Kodrat, Demak Lumban Tobing, Tiara Bunga Mayang Permata.

**Validation:** Handoko, Lisnawati Rachmadi.

**Visualization:** Handoko, I. Made Haryoga, Agustinus Gatot Dwiyono, Yoseph Adi Kristian.

**Writing – original draft:** Handoko.

**Writing – review & editing:** Handoko.

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
