## [Decision Letter · Decision Letter 0]

7 Feb 2020

PONE-D-19-33450

Tumor Microenvironment Predicts Local Tumor Extensiveness in PD-L1 Positive Nasopharyngeal Cancer

PLOS ONE

Dear Mr. Handoko,

Thank you for submitting your manuscript to PLOS ONE. After careful consideration, we feel that it has merit but does not fully meet PLOS ONE’s publication criteria as it currently stands. Therefore, we invite you to submit a revised version of the manuscript that addresses the points raised during the review process.

As you can see, both reviewers have made suggestions for minor changes to the manuscript.  We would request your consideration of these changes. 

We would appreciate receiving your revised manuscript by Mar 23 2020 11:59PM. To enhance the reproducibility of your results, we recommend that if applicable you deposit your laboratory protocols in protocols.io, where a protocol can be assigned its own identifier (DOI) such that it can be cited independently in the future. For instructions see: http://journals.plos.org/plosone/s/submission-guidelines#loc-laboratory-protocols

We look forward to receiving your revised manuscript.

Kind regards,

Salvatore V Pizzo

Academic Editor

PLOS ONE

Journal Requirements:

2. Please provide additional details regarding participant consent. In the ethics statement in the Methods and online submission information, please ensure that you have specified what type of consent you obtained (for instance, written or verbal). If your study included minors under age 18, state whether you obtained consent from parents or guardians. If the need for consent was waived by the ethics committee, please include this information.

Reviewers' comments:

Reviewer's Responses to Questions

**Comments to the Author**

1. Is the manuscript technically sound, and do the data support the conclusions?

Reviewer #1: Yes

Reviewer #2: Partly

2. Has the statistical analysis been performed appropriately and rigorously? 

Reviewer #1: Yes

Reviewer #2: Yes

3. Have the authors made all data underlying the findings in their manuscript fully available?

Reviewer #1: Yes

Reviewer #2: Yes

4. Is the manuscript presented in an intelligible fashion and written in standard English?

Reviewer #1: Yes

Reviewer #2: Yes

5. Review Comments to the Author

Reviewer #1: In this manuscript, the authors found that the PD-L1 was highly expressed in 25 NPC samples. They also revealed that denser local tumor infiltrating immune cells population have relatively much small local tumor volume. Thus, tumor infiltrating immune cells play an important role in tumor progression. The detection of tumor infiltrating immune cells may offer valuable prognostic information for NPC. This manuscript is well-written and the figures are well-presented. I only have one comment. I wonder if these NPC patents are receiving chemotherapy during this project. Chemotherapy or other therapies may have an impact on the distribution and density of tumor infiltrating immune cells. The authors should clarify this in the manuscript.

Reviewer #2: 1.Local tumor volume can be measured by CT or MRI scan exactly. There is no need to "predict" it. In this research the authors indicated the relationship between tumor volume and its microenviroment, changing the title into, for example, "Tumor microenvironment charactaristics in different tumor extensiveness in PD-L1 positive nasopharyngeal cancer" would more appropriate.

2.Did the enrolled patients received any treatment before biopsy? Medical treatment can influence the tumor progression and its microenviroment and further affect the final results.

3.In line 153

Did the enrolled patients have any hematological disease or infectious complications when taking peripheral blood count? These factors can affect the peripheral blood components and authors should illuminate this.

4.In line 174

It may be insufficient to use HE dye to evaluate tumor infiltrating immune cells level because it is too subjective. CD45 is widely expressed in all kinds of immune cells, so I suggest that CD45 staining is need to help evaluating the infiltrating level.

5.In line 187

"imaging selected" means "selected imaging" ?

6.PD-L1 is an important immunosupression molecular. So what is the relationship between PD-L1 expression and tumor infiltrating immune cells? In other kinds of cancers, PD-L1 often has higher expression in "hot tumor".

6. PLOS authors have the option to publish the peer review history of their article (what does this mean?). If published, this will include your full peer review and any attached files.

Reviewer #1: No

Reviewer #2: No

---

## [Author Response · Author response to Decision Letter 0]

14 Feb 2020

Reviewer #1: 

In this manuscript, the authors found that the PD-L1 was highly expressed in 25 NPC samples. They also revealed that denser local tumor infiltrating immune cells population have relatively much small local tumor volume. Thus, tumor infiltrating immune cells play an important role in tumor progression. The detection of tumor infiltrating immune cells may offer valuable prognostic information for NPC. This manuscript is well-written and the figures are well-presented. I only have one comment. I wonder if these NPC patents are receiving chemotherapy during this project. Chemotherapy or other therapies may have an impact on the distribution and density of tumor infiltrating immune cells. The authors should clarify this in the manuscript.

Thank you for the comment. The answer was no. All subjects included in this study did not have any prior cancer treatment. It’s true that some systemic treatments might impact tumor microenvironment. Thank you for pointing it out for us, now in our revised manuscript we have added a statement clarifying that issue.

Reviewer #2: 

1.Local tumor volume can be measured by CT or MRI scan exactly. There is no need to "predict" it. In this research the authors indicated the relationship between tumor volume and its microenviroment, changing the title into, for example, "Tumor microenvironment charactaristics in different tumor extensiveness in PD-L1 positive nasopharyngeal cancer" would more appropriate.

Thank you for the comment and suggestion. Yes it is true that we can obtain exact tumor volume by CR or MRI scan. Your suggestion “Tumor microenvironment characteristics in different tumor extensiveness in PD-L1 positive nasopharyngeal cancer" was great, however, the term “in different tumor extensiveness” could be connoted by some people as different disease stages. While actually in our study we used exact tumor volume. 

We used “predict” because we wanted to give the readers an upfront impression that tumor microenvironment has prognostic role when just glancing at our title. We refereed “predict” here as predicting role of tumor microenvironment. Therefore, we actually tend to stick to our original title. But we are still open to discussion if you have any other comments or ideas.

2.Did the enrolled patients received any treatment before biopsy? Medical treatment can influence the tumor progression and its microenviroment and further affect the final results.

No, all patients did not receive any prior treatment. We have revised and included a clear statement in our revised manuscript that all subjects enrolled in our study did not ever received any kind of treatment before.

3.In line 153

Did the enrolled patients have any hematological disease or infectious complications when taking peripheral blood count? These factors can affect the peripheral blood components and authors should illuminate this.

No, all patients enrolled did not have any hematological disease nor infections. The routine peripheral blood checks was part of our institution protocol. All patients would have the blood checked before biopsy and every week during treatment. Furthermore, the patient would not be biopsied if they had any kind of infection. We have also included a statement in our revised manuscript stating that all patients enrolled did not have any other severe diseases such hematological diseases or infectious diseases. 

4.In line 174

It may be insufficient to use HE dye to evaluate tumor infiltrating immune cells level because it is too subjective. CD45 is widely expressed in all kinds of immune cells, so I suggest that CD45 staining is need to help evaluating the infiltrating level.

Yes, we were aware about the subjectivity. Therefore, we have actually made a precautionary step by having 2 independent pathologists blindly reviewing the HE slides. Nevertheless, we are very happy with your suggestion. Next time we will add CD45 staining to help us tackle this subjectivity issue.

5.In line 187

"imaging selected" means "selected imaging" ?

Yes, we meant selected imaging. We have revised it in our revised manuscript. Thanks for point it out for us.

6.PD-L1 is an important immunosupression molecular. So what is the relationship between PD-L1 expression and tumor infiltrating immune cells? In other kinds of cancers, PD-L1 often has higher expression in "hot tumor".

There was no relationship found between PD-L1 expression and tumor infiltrating immune cells in our study. We were aware that some tumor did have relationship exist between those two. We have added a statement in our revised manuscript describing the absence of relationship between PD-L1 and tumor infiltrating immune cells in our NPC samples.

---

## [Decision Letter · Decision Letter 1]

2 Mar 2020

Tumor Microenvironment Predicts Local Tumor Extensiveness in PD-L1 Positive Nasopharyngeal Cancer

PONE-D-19-33450R1

Dear Dr. Handoko,

We are pleased to inform you that your manuscript has been judged scientifically suitable for publication and will be formally accepted for publication once it complies with all outstanding technical requirements.

With kind regards,

Salvatore V Pizzo

Academic Editor

PLOS ONE

Additional Editor Comments (optional):

Reviewers' comments:

Reviewer's Responses to Questions

**Comments to the Author**

1. If the authors have adequately addressed your comments raised in a previous round of review and you feel that this manuscript is now acceptable for publication, you may indicate that here to bypass the “Comments to the Author” section, enter your conflict of interest statement in the “Confidential to Editor” section, and submit your "Accept" recommendation.

Reviewer #1: All comments have been addressed

Reviewer #2: All comments have been addressed

2. Is the manuscript technically sound, and do the data support the conclusions?

Reviewer #1: Yes

Reviewer #2: Yes

3. Has the statistical analysis been performed appropriately and rigorously? 

Reviewer #1: Yes

Reviewer #2: Yes

4. Have the authors made all data underlying the findings in their manuscript fully available?

Reviewer #1: Yes

Reviewer #2: Yes

5. Is the manuscript presented in an intelligible fashion and written in standard English?

Reviewer #1: Yes

Reviewer #2: Yes

6. Review Comments to the Author

Reviewer #1: The authors have well addressed all my comments in the revised version of the manuscript. I have no further comment.

Reviewer #2: (No Response)

7. PLOS authors have the option to publish the peer review history of their article (what does this mean?). If published, this will include your full peer review and any attached files.

Reviewer #1: No

Reviewer #2: No

---

## [Editor Report · Acceptance letter]

6 Mar 2020

PONE-D-19-33450R1 

Tumor Microenvironment Predicts Local Tumor Extensiveness in PD-L1 Positive Nasopharyngeal Cancer 

Dear Dr. No Last Name:

I am pleased to inform you that your manuscript has been deemed suitable for publication in PLOS ONE. Congratulations! Your manuscript is now with our production department. 

With kind regards,

on behalf of

Dr. Salvatore V Pizzo 

Academic Editor

PLOS ONE